# Unsupervised Spiking Neural Network with Dynamic Learning of Inhibitory Neurons

**DOI:** 10.3390/s23167232

**Published:** 2023-08-17

**Authors:** Geunbo Yang, Wongyu Lee, Youjung Seo, Choongseop Lee, Woojoon Seok, Jongkil Park, Donggyu Sim, Cheolsoo Park

**Affiliations:** 1Department of Computer Engineering, Kwangwoon University, Seoul 01897, Republic of Korea; rmsqhwkd2@gmail.com (G.Y.); youjungseo0317@gmail.com (Y.S.); cndtjq97@gmail.com (C.L.); 2Department of Intelligent Information and Embedded Software Engineering, Kwangwoon University, Seoul 01897, Republic of Korea; eew1729@gmail.com (W.L.); swj20000@gmail.com (W.S.); 3Center for Neuromorphic Engineering, Korea Institute of Science and Technology (KIST), Seoul 02792, Republic of Korea; jongkil@kist.re.kr

**Keywords:** Bayesian inference, leaky integrate-and-fire model, spike timing-dependent plasticity, spiking neural network, unsupervised learning

## Abstract

A spiking neural network (SNN) is a type of artificial neural network that operates based on discrete spikes to process timing information, similar to the manner in which the human brain processes real-world problems. In this paper, we propose a new spiking neural network (SNN) based on conventional, biologically plausible paradigms, such as the leaky integrate-and-fire model, spike timing-dependent plasticity, and the adaptive spiking threshold, by suggesting new biological models; that is, dynamic inhibition weight change, a synaptic wiring method, and Bayesian inference. The proposed network is designed for image recognition tasks, which are frequently used to evaluate the performance of conventional deep neural networks. To manifest the bio-realistic neural architecture, the learning is unsupervised, and the inhibition weight is dynamically changed; this, in turn, affects the synaptic wiring method based on Hebbian learning and the neuronal population. In the inference phase, Bayesian inference successfully classifies the input digits by counting the spikes from the responding neurons. The experimental results demonstrate that the proposed biological model ensures a performance improvement compared with other biologically plausible SNN models.

## 1. Introduction

Neural networks that adopt biologically plausible models have recently become popular in numerous pattern recognition studies [1]. Popular machine learning (ML) models, such as deep neural networks (DNNs) [2,3,4], are inspired by biological neural networks, which mimic the information transmission among neurons in the human brain. However, a significant gap remains between the DNNs and the neuronal learning mechanisms found in biology in terms of the operation of the information transmission [5]. In contrast to the DNNs, which oversimplify the brain’s mechanism of transmitting signals to the adjacent neurons, the spiking neural networks (SNNs) mimic a variety of biological mechanisms based on the structural similarities between the brain and nervous system [6]. As fundamental spike-firing models for SNNs, the Hodgkin–Huxley [7] and Izhikevich [8] models formulate the biological mechanism of the neurons well. The leaky integrate-and-fire model (LIF) was proposed by Burkitt et al. [9] with a simpler biological architecture than the aforementioned models for better practical applicability. As a biologically plausible learning method, unsupervised spike timing-dependent plasticity (STDP) was proposed by Bi et al. [10,11] and adopted as a learning rule in several SNN architectures. Although it cannot exhibit biologically realistic long-term potentiation (LTP) or depression (LTD) in Bienenstock–Cooper–Munro behavior [10], encoded spike trains based on biological mechanisms, such as temporal or rate encoding, can reduce the difference between the classical LTP/LTD mechanisms and STDP [11]. The rate encoding mechanism is widely used in various SNN architectures, converting information into spikes based on the mean spike-firing rate of the stimulus. In temporal encoding, information is encoded with the precise spike timing corresponding to the stimulus intensity [12]. Both biological encoding methods demonstrate satisfactory classification performance with the STDP-based spiking neural networks  [13].

Although several SNNs designed based on biological models have been recently proposed, hybrid models of SNNs with DNNs have been suggested for better performance [14,15,16]. For example, the structure of convolutional neural networks (CNNs) [17] has been applied to SNNs, creating “Spiking CNNs” [18,19]. Even though these models had improved classification performance, they were slightly distant from bio-realistic neural networks. Moreover, other biophysical methods rely on supervised learning, where the neurons are provided with label information for classification problems [20]. Spike-Prop [21], which first used the backpropagation algorithm, showed that a feedforward SNN using error backpropagation could solve the nonlinear classification problem by solving the extended XOR problem in a time-coded network. Several studies have reported that error minimization based on label information is biologically plausible [22,23,24]. In contrast, other studies have argued that the mechanism of neurons learning input stimuli via minimizing the error between the predicted output and the correct answer does not seem to be biologically plausible [5,25,26].

For this reason, biologically plausible SNN models inspired by biological mechanisms have been proposed. Khacef et al. [27] proposed an unsupervised SNN based on a self-organizing map algorithm [28] mimicking the learning process of the visual cortex. Its network structure determines the firing neuron by selecting the smallest distance between the input stimulus and the weight of the neurons. A heterogeneous spiking neural network (H-SNN) proposed by Xueyuan et al. [29] analyzes the patterns of long-term and short-term memory using an STDP-based learning process. It performs the prediction of a moving object based on feedforward connections without a recursive structure. Compared with supervision-based SNNs, it yielded similar or better performance.

In this paper, we present a new SNN that imitates biologically plausible mechanisms in an unsupervised manner. The learning process in the human brain can be explained as follows: when a group of neurons are stimulated corresponding to an input, their synapses become strengthened as the learning proceeds. Thus, if the same type of stimulus is received repeatedly, the population of neurons that respond to the stimulus work together to fire spikes [30,31,32,33]. In the proposed SNN algorithm, the group of neurons creating the spikes corresponding to the specific stimulus are wired based on the synaptic wiring method. When a new input from the same category is fed into the network, the synaptically wired neurons create spikes together, which is how the algorithm learns a dataset. The basic architecture of the SNN is inspired by Diehl et al. [34], whose work is updated with the addition of the following models: inhibition weight updating of the inhibitory neurons, synaptic wiring among the neurons, and Bayesian inference based on a biologically plausible concept.

The proposed SNN in Figure 1 comprises three layers: the input neuron layer, first neuron layer, and second neuron layer. First, for the input data, handwritten digit images from the MNIST dataset [35] or handwritten letter images from the EMNIST dataset [36] are fed into the input neuron layer in the form of spikes using Poisson encoding. Next, the Poisson spike trains from the input neuron layer are delivered into the first neuron layer, where STDP works as a learning rule in a fully connected fashion. Third, the inhibition weight update method is applied between the neurons in the first and the second neuron layers by changing the values of the inhibition weight dynamically. Finally, the synaptic wiring of the neurons in the first neuron layer is performed. The neurons of the first and second neuron layers act either as presynaptic or postsynaptic neurons. If one neuron stimulates the following neuron, the first is called the presynaptic neuron and the second is called the postsynaptic neuron. A single neuron could have thousands of synaptic connections with many different postsynaptic neurons. Further, the neuronal connections are not static but vary over time [37]. The neurons that actively respond to the input data learn to fire spikes together once they are grouped together. The suggested SNN model aimed to improve classification performance with the MNIST and EMNIST datasets. After the learning process, Bayesian inference is applied to the final phase of the classification problem, which is also biologically plausible [38,39,40,41,42,43].

The remainder of this paper is structured as follows. Section 2 describes the architecture: the proposed layers and neuronal connections. Section 3 details the simulation results in comparison with another bio-plausible unsupervised SNN architecture. Section 4 presents an evaluation of the biological reasoning of our models. Section 5 concludes this paper.

## 2. Network Architecture

Python (version 3.7) and the BindsNET (version 0.3.0) simulator [44] were used to simulate the proposed SNN. The input images were obtained from the MNIST or EMNIST datasets, which contain pixel images of the digits 0–9 and the lower-case letters a–j.

To model the spiking activities in neurons, the LIF model [9] was applied to the network, in which the membrane voltage is described as
(1)$$\tau \frac{du}{dt}=-\left(u\left(t\right)-u_{rest}\right)+RI\left(t\right),$$
where $u_{rest}$ is the resting membrane potential of a neuron, I(t) is the constant input current that is activated corresponding to the spike, *R* is the resistance, and τ is the time constant. The parameter value of τ is 20 ms, $u_{rest}$ = −65 mV, *R* = 1, $\theta_{th}$ = −40 [44,45]. In this model, spike firing occurs when the potential reaches the membrane threshold, $\theta_{th}$. Once the potential reaches $\theta_{th}$, the neuron fires a spike, and then its potential value drops to $u_{rest}$. Subsequently, the neuron goes into a refractory state for 5 ms. During this period, the neuron cannot spike again and stays in its resting state ($u_{rest}$). The LIF model has been widely used owing to its ability to capture the intuitive properties of external input accumulating charge across a leaky cell membrane with a clear threshold [46,47,48]. Additionally, it is much simpler to implement in software and hardware systems than the previously proposed Hodgkin–Huxley model, which has a high computational cost [49].

### 2.1. Input Neuron Layer

The input neuron layer has a size of 784, which is the pixel size of MNIST and EMNIST images (28 × 28). Hence, each pixel of an input image is mapped onto each neuron in the input neuron layer. To present the input to the network, the image is encoded in the form of Poisson spike trains, which are fired depending on the intensity of the image pixels. The input firing rates are chosen between 0 and 127.5 Hz, designed to correspond to the maximum 255-pixel intensity of the images divided by 2 [34,50]. Additionally, the images are trained in grayscale such that there is no need to consider the multiple dimensions of the RGB channels.

#### STDP

STDP [51], the most representative biophysical model in SNNs, is applied for the weight update between the input neuron layer and the first neuron layer. STDP is defined as follows, considering $t_{pre}$ and $t_{post}$ as the time instances when the presynaptic and postsynaptic spikes occur:(2)$$\begin{array}{c} \Delta w_{+}=A_{+}\left(w\right)\exp(-\frac{\left|t_{post}-t_{pre}\right|}{\tau_{+}})\;,\;t_{pre}\leq t_{post}, \end{array}$$
(3)$$\begin{array}{c} \Delta w_{-}=A_{-}\left(w\right)\exp(-\frac{\left|t_{post}-t_{pre}\right|}{\tau_{-}})\;,\;t_{pre}>t_{post}, \end{array}$$
where $\Delta w_{+}$ is a positive weight change that occurs when the postsynaptic spike follows a presynaptic spike and $\Delta w_{-}$ is the negative weight change that occurs when a presynaptic spike follows a postsynaptic spike. $A_{\pm }\left(w\right)$ is the hyperparameter that adjusts the degree of weight update applied to the synapse. $\tau_{\pm }$ represents the time constants. The parameter values are as follows: $A_{+}=10^{-4}$, $A_{-}=10^{-2}$, $\tau_{\pm }=1.0$; these values were used in the studies by Saunders et al. [52] and BindsNET [44]. Once the MNIST input data are fed into the input neuron layer, the weights are instantaneously updated corresponding to the time instants, $t_{pre}$ and $t_{post}$.

### 2.2. First Neuron Layer

The input neuron layer receives the encoded spikes of the input image data and delivers the spikes to the first neuron layer with the STDP learning rule in a fully connected fashion. Next, the neurons in the first neuron layer are connected successively to the neurons in the second neuron layer in a one-to-one manner. The number of neurons in the first and second layers is identical and can be adjusted as a hyperparameter. The connection between the layers closely follows what Diehl et al. proposed [34].

In this regard, the order of the weight connection is important because it determines whether the neuron is excitatory or inhibitory. From a biological perspective, presynaptic and postsynaptic neurons communicate by transferring chemicals called neurotransmitters, which are represented as positive or negative weights in our proposed SNN model. A presynaptic neuron can be either excitatory or inhibitory with respect to its postsynaptic neuron, depending on which neurotransmitter it transmits [53]. If the presynaptic neuron induces an excitation of the postsynaptic neuron, the presynaptic neuron is called an excitatory neuron. In contrast, if the presynaptic neuron prevents the excitation of a postsynaptic neuron, the presynaptic neuron is called an inhibitory neuron [54]. Thus, in the proposed model, a presynaptic neuron becomes an excitatory neuron if it is wired with a positive weight toward the following neuron. Similarly, the neuron is an inhibitory neuron if it is connected with a negative weight to the following neuron. Figure 2 illustrates the method of defining an excitatory or inhibitory neuron based on the neuronal connection.

### 2.3. Second Neuron Layer

In the proposed SNN, the neurons in the first neuron layer are connected as presynaptic neurons to the neurons in the second neuron layer. Likewise, when the second neuron layer is connected back to the first neuron layer, the neurons in the second neuron layer become presynaptic neurons with respect to the neurons in the first neuron layer. To model these connections between the two layers, we introduce a matrix called a “weight matrix” in Figure 3. For example, in Figure 3, “+25” in the blue box denotes that F1 is connected to S1 with a positive weight of 25. Similarly, “0” means that there is no connection between the neurons, and “−50” represents that F3 is connected to S3 with a negative weight of 50.

In the initial setup, the synaptic connection from the first neuron layer to the second neuron layer occurs in a one-to-one fashion, where each neuron in the first neuron layer is connected to only one neuron in the other layer with either positive or negative weights. Based on the definition of excitatory and inhibitory neurons, the F1 neuron in Figure 3 is an excitatory neuron with regard to S1 since F1 is connected to S1 with +25 weight. In contrast, F3 acts as an inhibitory neuron for S3 owing to its connection with the weight of −50.

After the forward connection from the first neuron layer to the second neuron layer, a backward connection occurs, where the second neuron layer inhibits all of the neurons in the first neuron layer with a −50 inhibition weight except the ones connected in a one-to-one manner, which is called “lateral inhibition”. However, the static inhibition weight in this process could be problematic since the inhibition intensity of the biological inhibitory neuron is dynamic rather than static [55,56]. In this context, we propose a dynamically changing inhibition weight update model referred to as “inhibition weight update”.

#### 2.3.1. Inhibition Weight Update

The purpose of the inhibition weight update is to identify constantly firing neurons with an input label in the first neuron layer and increase their numbers of spike firings. In that respect, the maximum spiking neuron, which is selected under the condition of firing more than four spikes, is neglected in the inhibition weight update process because it is presumed to be fully trained for the input image. Since the spike-firing activity could be random if a neuron is not sufficiently trained, the heuristic threshold to select the most spiking neuron was set to four. The remaining neurons update their inhibition weights, $W_{SF}$, based on the following rule:(4)$$W_{SF}=W_{SF}+L\left(S_{n}-S_{n-1}\right),$$
where $W_{SF}$ is the weight from the inhibitory neuron corresponding to the most spiking excitatory neuron in response to the other firing excitatory neurons. $S_{n}$ and $S_{n-1}$ are the total numbers of spikes generated in the previous and current spike trains from one neuron. Note that this equation is based on the gradient descent algorithm [57].

Based on Equation (Equation 4), Algorithm 1 elaborates the pseudo code of the inhibition weight update, where the initial weight of $W_{SF}$ is −50. The learning rates for the weight updates $L_{1}$ and $L_{2}$ were chosen heuristically as 0.5 and 0.07. Experimental observation showed that the event of spike firing becomes less frequent as the training continues. Thus, the learning rate for the negative update $L_{2}$ was set lower than the positive update $L_{1}$.
**Algorithm 1** Inhibition Weight Update and Synaptic Wiring**Require:** Weight from inhibition neuron in relation to excitatory layer $W_{SF}$, synaptic wiring threshold $\vartheta_{inhibit}$, inhibition weight learning rate $L_{1}$ and $L_{2}$, maximum firing neuron $N_{m}$1:Set *W*_*SF*_ = −50, $\vartheta_{inhibit}$ = −38, *L*_1_ = 0.5, *L*_2_ = 0.072:// Inhibition weight update3:*N*_*m*_ = most active neuron among reacted excitatory neurons with Sn ≥ 44:**for** ∀ reacted excitatory neurons based on the *N*_*m*_ and $W_{SF}<0$
**do**5:    **if** $S_{n}>S_{n-1}$ **then**6:        $W_{SF}\leftarrow W_{SF}+L_{1}\left(S_{n}-S_{n-1}\right)$7:    **else**8:        $W_{SF}\leftarrow W_{SF}+L_{2}\left(S_{n}-S_{n-1}\right)$9:    **end if**10:    // Synaptic wiring11:    **if** $W_{SF}<0$ and $W_{SF}>\vartheta_{inhibit}$ **then**12:        $W_{SF}=+25$13:    **end if**14:**end for**

#### 2.3.2. Synaptic Wiring

This section introduces a new synaptic wiring method, as illustrated in Figure 4. The synaptic wiring scheme is based on the synaptic plasticity of the biological system, widely known as Hebb’s rule—“repeatedly firing neurons are wired together” [58]—and on the population activity of the neurons [32,33,59,60]. The synaptic plasticity rule states that the synaptic connection already created between the presynaptic and postsynaptic neurons can be modified as the learning progresses [58]. Additionally, when neurons receive an environmental stimulus, it is processed by the activity of the population of neurons rather than a single neuron, and this is referred to as the population activity of neurons [32,33,59,60].

The proposed model aims to mimic these biological mechanisms mentioned above. At first, the synaptic plasticity of the inhibitory neurons in the second neuron layer is implemented. The inhibitory neurons in the second neuron layer, which are connected to the postsynaptic neurons in the first neuron layer with the weight −50, go through the process of the weight update. Once the electrical potential in the inhibitory neuron reaches the threshold value, the inhibitory connection from the presynaptic neuron in the second neuron layer to the postsynaptic neuron in the first neuron layer is changed to the excitatory connection. Second, the mechanism of the neural population activity is implemented; that is, once the synaptic connection occurs, the multiple postsynaptic neurons wired in the first neuron layer fire the spikes together in response to the input.

The first step of this synaptic wiring process involves updating the inhibition weights between the second neuron layer and the first one, as described in Algorithm 1. Next, the synaptic wiring occurs between the maximum spiking neuron in the first layer and the neurons whose inhibition weight in relation to the second-layer neurons is increased over the threshold value, −38, and changed to +25. This is elaborated in the “Synaptic wiring” part of Algorithm 1, where $\vartheta_{inhibit}$ (−38) denotes the threshold of the increased inhibition weight in relation to the inhibition weight. Consequently, neurons wired together in the first neuron layer are encouraged to fire more spikes through the involvement of multiple neurons rather than a single one. After this, the group of wired spiking neurons are assigned to the given input label [61].

Figure 4 depicts the process of the synaptic wiring of the neurons, where step one represents the one-to-one connection of the neurons from the first neuron layer to those from the second neuron layer. Specifically, the third neuron in the first neuron layer is connected to the third neuron in the second neuron layer with a weight of +25, represented as $W_{FS}$ (3, 3) = 25. The connections from excitatory neuron layer to the inhibitory neuron layer have a fixed weight since they undergo no weight update. The role of these connections is to check whether the spike from the excitatory neuron that corresponds to the inhibitory neuron is fired. Step two in Figure 4 illustrates the synaptic wiring of the 3rd and 24th neurons in the first neuron layer through the second neuron layer. Suppose that the third neuron in the first neuron layer is currently the maximum spiking neuron. Once the weight of the inhibition transmitted from the second neuron layer to the third neuron in the first neuron layer increases over the threshold value −38, it is forced to change to +25—that is, $W_{SF}$ (3, 24) = 25—to implement synaptic wiring of 3rd and 24th neurons. Consequently, the 3rd and 24th neurons in the first neuron layer are encouraged to fire spikes together, responding to the same input as a result of the synaptic wiring. This neuronal connection is rooted in the neurobiological system [61]. The weight −38 is empirically selected based on the classification performance depending on $W_{SF}$ in Equation (Equation 4). In Table 1, the results from testing using various weights with 1600 neurons are presented and, as can be seen, −38 was the optimal weight for producing the best result.

In an unsupervised scheme, the spike firings of the neurons are fully dependent on the input data. In other words, there is no guarantee that every neuron has an equal chance of firing in response to the input data. Thus, the spike firing of the excitatory neurons should be adjusted. Additionally, to avoid certain neurons dominating the spike firing, an adaptive membrane threshold model can be applied [62,63]. Using this model, the inhomogeneity of the spike firing between the neurons can be controlled, resulting in the homeostasis of every neuron in firing spikes.

### 2.4. Inference

Biophysical research suggests that the human brain employs Bayesian inference when making decisions or processing external stimuli [38]. We also attempted to apply the Bayesian inference algorithm to predict the class label in our SNN, similar to how the human brain interprets the outer environment. After the inhibition weight update and the synaptic wiring take place, the spikes fired from the neurons are counted to predict the class label. To derive the Bayesian inference for the neurons, a proper probability density function (PDF) must be selected in advance. In this simulation, the Poisson PDF was adapted based on the equation given below:(5)$$P\left(s\;|c_{i},n_{j}\right)=\frac{\lambda^{s}e^{-\lambda }}{s!}$$
where *s* denotes the number of spikes, $c_{i}$ denotes the *i*th class label image, $n_{j}$ denotes the *j*th excitatory neuron, and λ denotes the expected value of the spikes fired by $n_{j}$.

First, the neurons in the SNN generated independent spike firings for 350 ms. The spike firings of the neurons were counted for 350 ms, and the collected spikes were re-modeled with the Poisson distribution based on the class and order. The spike firing duration was 350 ms; this duration was used in BindsNET [44], which was implemented based on the study by Diehl et al. [34]. This firing duration can be explained in a biologically plausible way. For example, when our optic cells detect an object, the visual neurons generate spikes for 350 ms [64,65].

The Poisson distribution was believed to be suitable for our model because the distribution represents the occurrence of independent events in a given window, such as the spike events [66]. During the training process, a prior probability, $P(c_{i}|n_{j})$, was calculated and utilized in the testing process to derive the posterior in Equation (Equation 6) using the likelihood, $P(s|c_{i},n_{j})$, in Equation (Equation 5).
(6)$$P\left(c_{i}|s,n_{j}\right)=\frac{P\left(c_{i}\;|n_{j}\right)P\left(s\;|c_{i},n_{j}\right)}{P(s\;|n_{j})}\propto P\left(c_{i}\;|n_{j}\right)P\left(s\;|c_{i},n_{j}\right)$$

Since the chance of each neuron firing spikes is assumed to be equal, the denominator term in Equation (Equation 6) can be ruled out. To derive the probability of the classes for each neuron, the joint probability of each neuron and the class label can be estimated as follows:(7)$$P\left(c_{i}|s,n_{j}\right)P\left(n_{j}|s\right)=P\left(n_{j},c_{i}|s\right)$$
(8)$$\sum_{j}P\left(n_{j},c_{i}|s\right)=P\left(c_{i}|s\right)$$

The class label with the highest probability of $P(c_{i}|s)$ can be taken as the final estimation of the inference, as presented below:(9)$$\underset{i}{\mathrm{argmax}}P\left(c_{i}|s\right)$$

## 3. Results

To evaluate the proposed SNN model, different sizes were selected for the dataset and the model was trained depending on the number of excitatory and inhibitory neurons. The size of the training data was chosen such that the best accuracy could be obtained. However, 10,000 and 4000 samples from the MNIST and EMNIST datasets, respectively, were used with the test data for all the cases. The accuracy was calculated based on the average from 10 repetitions in the test cases. The testing results for the MNIST dataset are summarized in Table 2. The proposed model achieved classification accuracy of 84.48%, 87.25%, 90.05%, 90.72%, 92.30%, and 92.21% for 200, 400, 900, 1100, 1600, and 3000 neurons, respectively. The testing results for the EMNIST dataset are shown in Table 3. The proposed model achieved classification accuracy of 81.20%, 84.45%, 87.35%, 88.20%, 88.70%, and 90.04% for 200, 400, 900, 1100, 1600, and 3000 neurons, respectively. Note that the number of synaptic wirings increased with the number of neurons.

Table 4 shows the results for the proposed model using the MNIST and EMNIST datasets compared with other benchmarked models, for which the results for the model by Diehl et al. [34] and the unsupervised version of the sym-STDP-based SNN model proposed by Hao et al. [67] were selected. Both models are biologically plausible and implemented based on the BindsNET (version 0.3.0) simulator. For the benchmark test with the MNIST dataset, Diehl et al.’s model [34] achieved 82.68%, 77.70%, 88.14%, 87.92%, 87.85%, and 89.77% and Hao et al.’s model [67] yielded 78.56%, 87.18%, 86.85%, 87.44%, 88.60%, and 90.15% with 200, 400, 900, 1100, 1600, and 3000 neurons, respectively. Using the EMNIST dataset, Diehl et al.’s model [34] achieved 78.08%, 82.01%, 83.83%, 85.25%, 87.90%, and 89.77% and Hao et al.’s model [67] yielded 78.56%, 87.18%, 86.85%, 87.44%, 88.60%, and 90.15%, respectively.

Figure 5 displays the performances of the proposed and conventional models using different numbers of neurons (6000, 10,000, 25,000, 30,000, 75,000, and 90,000) for the training dataset. The proposed SNN model achieved the best classification accuracy of 92.30% using 1600 neurons. As a result, it was found that, for all the numbers of neurons, our model outperformed the bio-plausible SNN model proposed by Diehl et al. [34].

Figure 6 depicts the membrane potentials of the neurons in the first neuron layer, where the different colored signals represent the membrane potentials of the the different neurons. Among these neurons, only those producing membrane potentials higher than those of the spiking threshold fire spikes, which is represented by the arrows on top of each image in Figure 6. After the spike firing, all of the neurons, including the firing neurons, go to their resting states with low voltage potential and repeat the same process again after the refractory phase [9].

Figure 6a,b show the firing activities among the neurons without the synaptic wiring process. In Figure 6a, a single neuron can reach the firing threshold to generate spikes, as indicated with green arrows, while in Figure 6b, the neurons depicted in gray and blue can reach the threshold to fire the spikes alternately, competing with each other. For the final decision about how to classify the input image, only the neuron firing the greatest number of spikes is considered in the 350 ms window. Thus, in the simulation shown in Figure 6b, only the gray neuron, which was the maximum spiking neuron, was counted for the accuracy evaluation. However, along with the gray neuron, the blue one could be considered as a neuron that responds to the same input since the blue neuron also fires as many spikes as the gray neuron generates. To count the number of spike firings of both the gray and blue neurons, synaptic wiring was proposed to wire the two neurons so that they fire spikes together, as can be seen in Figure 6c. In Figure 6c, as the arrow indicates, the brown and orange neurons fire together owing to the synaptic wiring during the learning process of the proposed model. For the final decision process based on the Bayesian inference, the spikes of both the neurons are counted. By doing this, significantly spiking neurons, such as the blue neuron in Figure 6b, are not ignored in the final inference step.

## 4. Discussion

The proposed two-layer SNN architecture was carefully designed based on the biological plausibility. The following sections address the biological interpretation of the SNN model proposed in this study.

### 4.1. Inhibitory Neurons

In biological neural networks, the excitatory effect of a neuron is offset by the inhibitory neurons by dynamically changing the inhibition weights, thereby ensuring that the excitation of the neurons is controlled [68]. In contrast to the conventional SNN model proposed by Diehl et al. [34], our model induces spike firing from the excitatory neurons belonging to the same group using the synaptic wiring algorithm, applying the biological concept of the neuronal population activity to the inhibitory neuron layer. As a result, the two different forces between the excitatory and inhibitory neurons work together to create a rhythmic force. This force between the neurons is crucial for the brain to function smoothly as an information carrier [68]. Therefore, the dynamically changing inhibition weight between the inhibitory neuron and the excitatory neuron is critically important to produce the aforementioned behavior [69]. The proposed algorithm models the biological function with the implementation of the dynamic update of the inhibition weight.

### 4.2. Synapse

To interpret the creation of the synaptic connections among neurons, Hebb’s postulate, which states that “cells that repeatedly fire together, wire together”, is widely accepted in the field of neuroscience and related fields [70]. The proposed SNN model is based on this postulate, as it implements synaptic wiring among neurons. In examining Hebb’s postulate, two important aspects of Hebb’s theory have been noted [61]. First, the learning process for the neurons takes place “locally”. For example, if two neurons, *A* and *B*, are connected with each other, the other neurons may not be considered. Thus, the proposed model only considers the neuronal connection between two neurons; that is, the target neuron and the maximum spiking neuron in the first neuron layer. Second, the neurons involved in the connection must fire simultaneously. The proposed architecture also models the synaptic connection of the neurons firing simultaneously, given the data. Based on these aspects, it can be claimed that the proposed SNN is biologically plausible with regard to mimicking the human brain.

### 4.3. Biological Inference

During the testing process, synaptically connected neurons actively fire together, and the maximum spiking neuron is represented by a group of neurons rather than a single neuron. For the final inference of this neuron group, an additional biologically inspired inference method called Bayesian inference was applied. When humans and other animals are put in a situation where they need to make decisions and control movement with insufficient information, their brains tend to rely on prior knowledge in addition to current observations [39,40,41,42,43].

To implement this model, an appropriate PDF was first chosen (see Equation (Equation 5)). In the biological context, Poisson distribution is frequently used to explain the irregular timing of neuron spike firings [71]. Thus, we applied the Poisson distribution in calculating the probability of the spike firings of each neuron in relation to a given input. To verify this method, it was compared to another popular probability distribution; that is, the Gaussian distribution. The comparison of the two models is shown in Table 5, where the results clearly demonstrate that the Poisson PDF-based model outperformed the other. As presumed, the Poisson PDF is suitable for the prediction of spike events occurring in a given interval of time [66].

Figure 7 illustrates an example of spiking responses from a single neuron trained with the class-one dataset. The x-axis represents the number of spikes, and the y-axis represents the probability of the occurrence of each spike. Note that the overall spike firing distribution resembles the Poisson distribution.

## 5. Conclusions

In this work, we introduced a biologically plausible SNN model based on the synaptic wiring method and Bayesian inference. The former approach was implemented by dynamically changing the inhibition weight update, and the latter was based on the statistical properties of neuronal spike firing. The SNN model was designed mainly based on the neuroscientific phenomena of the human brain. The algorithms we developed improved the biological motivation for the lateral inhibition in the inhibitory neuron layer, as well as the inference, when compared to the conventional biological SNN model. In the experiment, the proposed SNN structure achieved high accuracy, outperforming other biologically plausible SNN models. The formation of a neuronal population through the inhibitory weight update process induced population firing, which led to a specific output neuron being more responsive to a specific input pattern. Furthermore, we proposed a Bayesian inference method inspired by biological mechanisms. The conventional inference method just uses the summation of spikes from each output neuron. The proposed inference method is a more probabilistic approach compared to the conventional inference method. In conclusion, the two proposed algorithms, synaptic wiring with inhibitory weight updating and Bayesian inference, led to performance improvement. Our model also showed superior performance with both a small number of neurons and a large number of neurons.

## Figures and Tables

**Figure 1 sensors-23-07232-f001:**
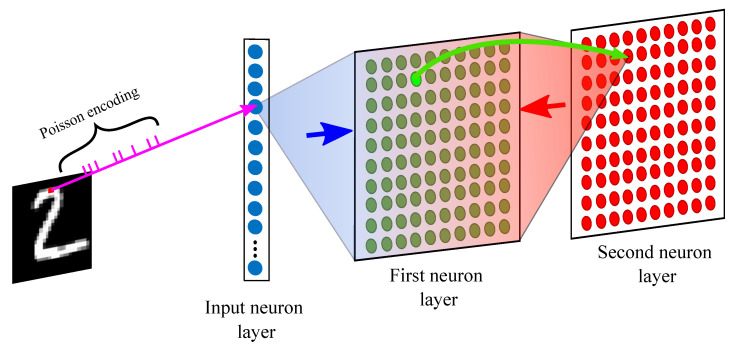
Each Poisson-encoded sample in the input neuron layer is connected to the neurons in the first neuron layer in a fully connected fashion (blue arrow). Next, the neurons in the first neuron layer are connected one-by-one to the neurons in the following layer, the second neuron layer (green arrow). Finally, the neurons in the second neuron layer inhibit the neurons in the first neuron layer, except the ones that have one-to-one connections (red arrow), which is called “lateral inhibition”.

**Figure 2 sensors-23-07232-f002:**
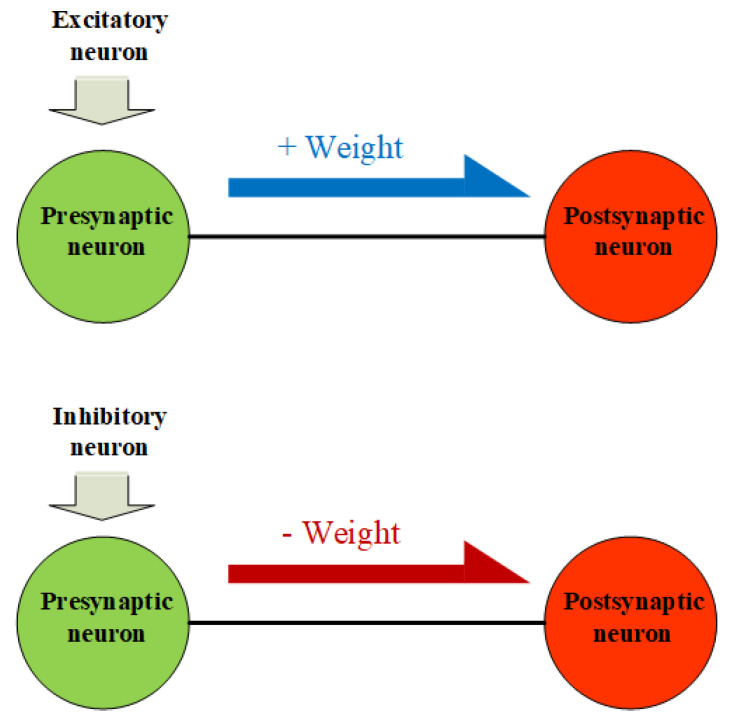
Definition of excitatory and inhibitory neurons based on the sign of the connection weight.

**Figure 3 sensors-23-07232-f003:**
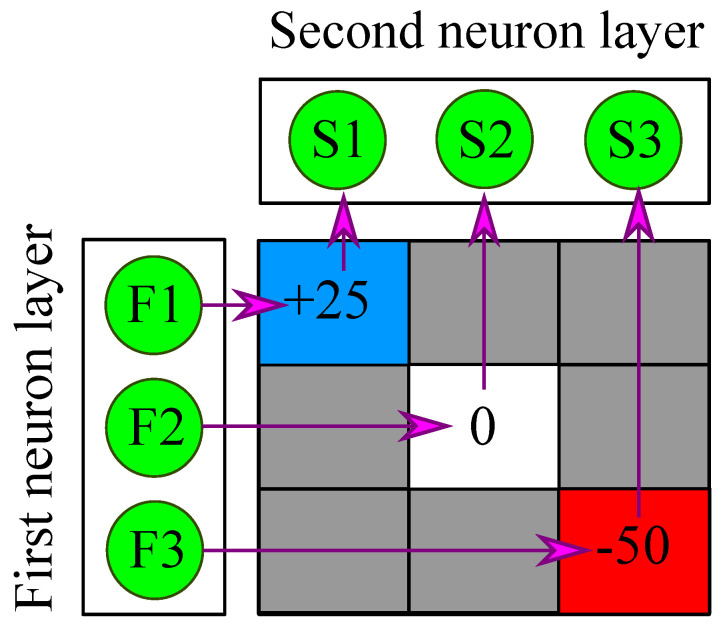
Weight matrix.

**Figure 4 sensors-23-07232-f004:**
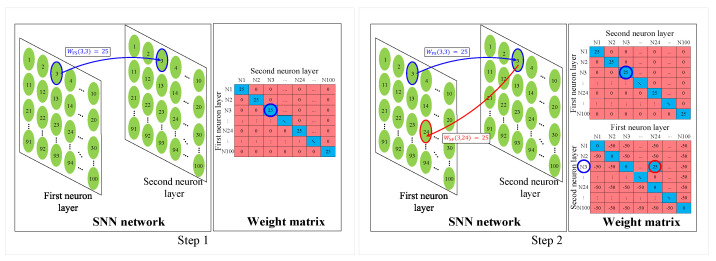
Synaptic wiring process. Step one: one-to-one mapping of the initial setting from the first neuron layer to the second neuron layer, $W_{FS}$(3, 3) = 25. Step two: the synaptic wiring between the 3rd neuron and the 24th neuron in the first neuron layer through the second neuron layer.

**Figure 5 sensors-23-07232-f005:**
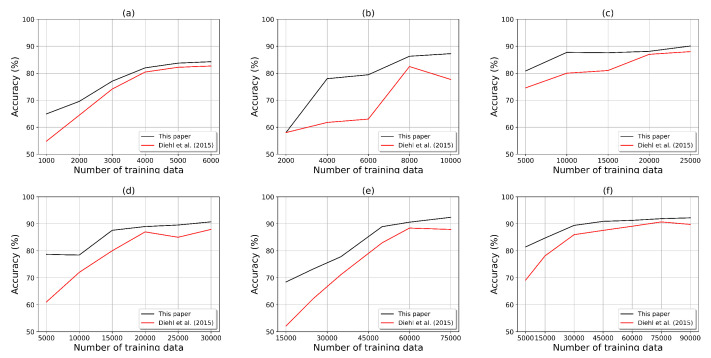
The test results when applying our model to the MNIST dataset in comparison with the conventional bio-plausible SNN model [34] based on the number of neurons: 200 (**a**), 400 (**b**), 900 (**c**), 1100 (**d**), 1600 (**e**), and 3000 (**f**).

**Figure 6 sensors-23-07232-f006:**
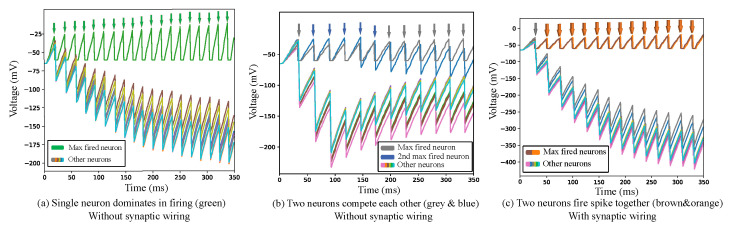
Membrane potentials of the neurons in the first neuron layer. The top arrows indicate the spike firings of a neuron after the potential reaches the firing threshold. Note that the multiple neurons fire spikes simultaneously with the synaptic wiring in (**c**).

**Figure 7 sensors-23-07232-f007:**
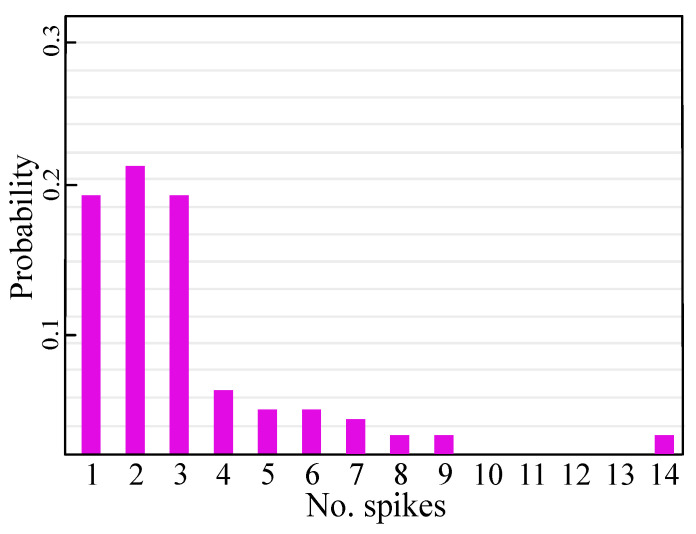
Firing activities of a single neuron for different input classes.

**Table 1 sensors-23-07232-t001:** Classification accuracy with the MNIST dataset with different threshold values.

Weight, $W_{\mathrm{SF}}$	Accuracy (%, 1600 Neurons)
−36	88.51
−37	88.20
−38	92.30
−39	90.15
−40	91.25

**Table 2 sensors-23-07232-t002:** MNIST dataset accuracy with varying amounts of excitatory and inhibitory neurons.

No. of Neurons	No. of Training Data Samples *	Accuracy (%)	No. of Synaptic Wirings
200	6000	84.48 ± 0.17	23
400	10,000	87.25 ± 0.15	65
900	25,000	90.05 ± 0.13	112
1100	30,000	90.72 ± 0.13	124
1600	75,000	92.30 ± 0.14	241
3000	90,000	92.21 ± 0.10	273

* The number of training data samples used to achieve the saturated accuracy.

**Table 3 sensors-23-07232-t003:** EMNIST dataset accuracy with varying amounts of excitatory and inhibitory neurons.

No. of Neurons	No. of Training Data Samples *	Accuracy (%)	No. of Synaptic Wirings
200	20,000	81.20 ± 0.13	15
400	44,000	84.45 ± 0.15	56
900	90,000	87.35 ± 0.15	189
1100	96,000	88.20 ± 0.14	276
1600	102,000	88.70 ± 0.19	387
3000	114,000	90.04 ± 0.20	392

* The number of training data samples used to achieve the saturated accuracy.

**Table 4 sensors-23-07232-t004:** Benchmark results comparing the proposed model with others using the same MNIST and EMNIST datasets.

No. of Neurons	Accuracy (%) for the MNIST Dataset			Accuracy (%) for the EMNIST Dataset		
	Proposed Model	Diehl et al. (2015) [34]	Hao et al. (2020) [67]	Proposed Model	Diehl et al. (2015) [34]	Hao et al. (2020) [67]
200	**84.48 ± 0.17**	82.68 ± 0.18	78.56 ± 0.22	**81.20 ± 0.13**	78.08 ± 0.11	77.78 ± 0.12
400	**87.25 ± 0.15**	77.70 ± 0.20	87.18 ± 0.12	**84.45 ± 0.15**	82.01 ± 0.13	81.25 ± 0.11
900	**90.05 ± 0.13**	88.14 ± 0.17	86.85 ± 0.21	**87.35 ± 0.15**	83.83 ± 0.12	84.50 ± 0.14
1100	**90.72 ± 0.13**	87.92 ± 0.15	87.44 ± 0.11	**88.20 ± 0.14**	85.25 ± 0.13	86.17 ± 0.11
1600	**92.30 ± 0.14**	87.85 ± 0.14	88.60 ± 0.18	**88.70 ± 0.19**	87.90 ± 0.13	87.60 ± 0.12
3000	**92.21 ± 0.10**	89.77 ± 0.12	90.15 ± 0.10	**90.04 ± 0.20**	87.02 ± 0.11	87.75 ± 0.14

**Table 5 sensors-23-07232-t005:** Performances of the Bayesian inference models based on Gaussian and Poisson PDFs.

Neurons	Training Samples	Accuracy (Gaussian)	Accuracy (Poisson)
200	6000	50.74 ± 0.35	84.48 ± 0.17
400	10,000	61.90 ± 0.32	87.25 ± 0.15
900	25,000	73.99 ± 0.20	90.05 ± 0.13
1100	30,000	77.24 ± 0.30	90.72 ± 0.13
1600	75,000	80.65 ± 0.33	92.30 ± 0.14
3000	90,000	83.26 ± 0.27	92.21 ± 0.10

## Data Availability

No new data were created in this study.

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
