# Peer review of "Unsupervised Spiking Neural Network with Dynamic Learning of Inhibitory Neurons"

_sensors, 2023, doi:10.3390/s23167232_

Round 1

Reviewer 1 Report

The authors propose a new spiking neural network with dynamic inhibition weight change designed for image recognition tasks. The paper has the following drawbacks.

1. In the Abstract, “the experimental results demonstrate that the proposed model ensures the performance improvement compared with the biologically plausible SNN models.” There is a misconception in this statement. The proposed model improves performance compared with the biologically plausible SNN models, so is not proposed model biologically plausible? I think this proposed SNN is also biologically plausible.

2. For the proposed SNN, only the number of neurons in the input layer is given, and the number of neurons in the other two layers should also be given clearly.

3. In the Results section, the authors carry on the contrast experiment based on different the number of neurons, the number of neurons refers to which layer of neurons.

4. Tables 2 and 3 have two variables (the number of neurons and the number of training data). Thus, it is not clear which variable influenced the result.

5. Table 4 expresses the same content as Figs. 5 and 6. Thus, Figs. 5 and 6 are unnecessary.

6. The two methods (Diehl et al. [31] and Hao et al. [63]) being compared need to be introduced to make reader understand easily. Besides dynamic inhibition weight change, whether the other factors of the two methods are consistent with the proposed method, such as neuron models and neuron connections. If yes, that would be OK. If not, the author needs to add a simulation to compare with the method without dynamic suppression weights to demonstrate the effectiveness of the proposed method.

7. The discussion needs to indicate why the proposed method is better than the other methods, rather than only showing the advantages of the proposed method.

8. The description of Algorithm 1 is incomplete.

Based on the above drawbacks, authors should prepare a major revision.

Minor editing of English language required

Reviewer 2 Report

The paper introduces a new Spiking Neural Network (SNN) that mimics the human brain's processing through discrete spikes. It incorporates biologically plausible principles and is designed for image recognition tasks. The SNN's unsupervised learning, dynamic inhibition weight change, synaptic wiring method, and Bayesian inference lead to improved performance compared to other SNN models.

This work is highly relevant in the field of computational neuroscience and its applied areas. It meets all the criteria of the journal and can be confidently recommended for acceptance.

Author Response

Response to Reviewer 1 Comments:

The paper introduces a new Spiking Neural Network (SNN) that mimics the human brain's processing through discrete spikes. It incorporates biologically plausible principles and is designed for image recognition tasks. The SNN's unsupervised learning, dynamic inhibition weight change, synaptic wiring method, and Bayesian inference lead to improved performance compared to other SNN models.

This work is highly relevant in the field of computational neuroscience and its applied areas. It meets all the criteria of the journal and can be confidently recommended for acceptance.

Author response:  I appreciate your review and comments.

Reviewer 3 Report

The manuscript develops a new biologically plausible spiking neural network (SNN) based on dynamic inhibition weight change, synaptic wiring method, and Bayesian inference. The basic architecture of this SNN is based on an existing work by Diehl et al., and uses the conventional spike-timing dependent plasticity (STDP) as the learning rule. The performance of this SNN is evaluated in image recognition tasks, using benchmark datasets including handwritten digits and letters and compared to existing biologically inspired SNN frameworks. Extending existing bioinspired SNNs to allow dynamic connections between layers, grouping neurons based on their correlated activity, and using it to design a Bayesian classifier shows the effectiveness of the proposed approach in improving classification performance over existing models with similar network and training data size.

The rigor of analytical and experimental studies is powerful. The write-up is well-organized and clear. The study presents the feasibility and potential of implementing biological principles for improving SNNs. 

Comments:

Page 4: “… where ∆wis a positive weight change that occurs when the postsynaptic spike follows a presynaptic spike and ∆w− is the negative weight change that occurs when a presynaptic spike follows a presynaptic <postsynaptic> spike.” 

Page 5: The heuristic threshold of 4 spikes for distinguishing trained vs untrained neurons may be a reasonable strategy for this preliminary demonstration of the new SNN; however, it makes more sense if this criterion can follow the specifics of data or other network parameters. For example, perhaps a trained neuron shows a more stable weight over learning iterations while an untrained neuron shows more variability in its weight over learning iterations, a criterion that can be used to differentiate them in a more data-specific manner and so improve the future versions of this model.

Page 8: Assuming a constant spike rate over a long period of 350 ms is not biophysically reasonable. Although it shows working for this preliminary SNN, the future versions can benefit from more realistic models of inhomogeneous Poisson spiking process, or response latency and decaying dynamics. 

Page 9: In addition to overall accuracy, a demonstration or discussion on what aspects of the classification task were improved (e.g., greater improvement for certain input patterns or handling certain distortions in the input) could have been informative. 
